# Function of Innate Lymphoid Cells in Periodontal Tissue Homeostasis: A Narrative Review

**DOI:** 10.3390/ijms24076099

**Published:** 2023-03-23

**Authors:** Zhiyu Ma, Jinsong Wang, Lei Hu, Songlin Wang

**Affiliations:** 1Beijing Laboratory of Oral Health, School of Basic Medicine, School of Stomatology, Capital Medical University, Beijing 100050, China; 2Department of Biochemistry and Molecular Biology, School of Basic Medicine, Capital Medical University Beijing 100070, China; 3Department of Prosthodontics, School of Stomatology, Capital Medical University, Beijing 100050, China; 4Immunology Research Center for Oral and Systemic Health, Beijing Friendship Hospital, Capital Medical University, Beijing 100070, China; 5Laboratory for Oral and General Health Integration and Translation, Beijing Tiantan Hospital, Capital Medical University, Beijing 100070, China; 6Research Unit of Tooth Development and Regeneration, Chinese Academy of Medical Sciences, Beijing 100700, China

**Keywords:** innate immunity, periodontitis (PD), innate lymphoid cells (ILC)

## Abstract

Periodontitis is an irreversible inflammatory response that occurs in periodontal tissues. Given the size and diversity of natural flora in the oral mucosa, host immunity must strike a balance between pathogen identification and a complicated system of tolerance. The innate immune system, which includes innate lymphoid cells (ILCs), certainly plays a crucial role in regulating this homeostasis because pathogens are quickly recognized and responded to. ILCs are a recently discovered category of tissue-resident lymphocytes that lack adaptive antigen receptors. ILCs are found in both lymphoid and non-lymphoid organs and are particularly prevalent at mucosal barrier surfaces, where they control inflammatory response and homeostasis. Recent studies have shown that ILCs are important players in periodontitis; however, the mechanisms that govern the innate immune response in periodontitis still require further investigation. This review focuses on the intricate crosstalk between ILCs and the microenvironment in periodontal tissue homeostasis, with the purpose of regulating or improving immune responses in periodontitis prevention and therapy.

## 1. Introduction

Periodontitis, a multifactorial disease, is associated with dysbiosis of the oral microbiota, aberrant inflammatory immune responses, and genetic predisposition. Microbial dysbiosis, initial oral-soft-tissue inflammation, and innate and adaptive immunity work together to protect periodontal tissue throughout the immune response, involving inflammation, regression, and healing. Although the adaptive immune system has been widely researched in the pathogenesis of periodontitis, the innate immune system in general and innate lymphoid cells, in particular have, received less attention.

Innate lymphoid cells (ILCs) are a heterogeneous population of newly discovered lymphoid cells that have recently been identified. They play central roles within mucosal barrier function, microbiome homeostasis, and control of inflammation in the early defense stage. ILCs are divided into five distinct ILC subsets, namely helper ILCs (i.e., ILC1s, ILC2s, ILC3s), natural killer (NK) cells, and lymphoid tissue inducer (LTi) cells, which are characterized by the production of different cytokines and functions through signature cytokines and transcription factors involved in their development [1,2,3]. ILC1s contributes to protective immune response directed against intracellular bacteria or viruses, as well as tumors through interferon-γ (IFN) [4]; ILC2s contribute to response to infections with extracellular parasites or microbes after being stimulated by allergens through interleukin (IL)-5, IL-9, and IL-13 [5]; ILC3s act against extracellular microorganisms (i.e., bacteria and fungi) through IL-22 and IL-17 [3,6]; and LTi cells play a crucial role in lymphoid tissue development during fetal life, and they may also help repair tissue in secondary lymphoid organs in adults. 

ILCs in different organs and tissues exhibit different immunophenotypes and functions, adapting to particular tissue microenvironments [7]. Research in the past decade has increased our comprehension of how ILCs affects the tumor and inflammation microenvironment. NK-cell-based immunotherapy has produced immune checkpoint inhibitors (ICIs) for the production of ILCs, and these molecules may regulate ILCs to control their activity in tumor microenvironments [8]. Although ILCs in health and chronic inflammatory diseases have been explored and their role in some chronic diseases, such as allergy and inflammatory bowel disease (IBD), have been proposed in recent years, the evidence from animal studies is far from conclusive. This study provides an overview of the phenotypic and developmental origin of ILC subsets and discusses what has come to be recognized as the functions of ILCs in periodontal tissue.

## 2. Innate Lymphoid Cells

Innate lymphoid cells (ILCs) can be divided into helper ILCs and cytotoxic ILCs (natural killer [NK] cells) based on cytokines and transcription factors [9] (Table 1).There are three distinct subpopulations of helper ILCs: ILC1s are T-bet-dependent and secrete interferon-γ (IFN-γ); ILC2s are dependent on GATA3 and secrete type 2 cytokines, such as interleukin (IL)-5 and IL-13; and ILC3s are dependent on retinoic acid receptor-related orphan receptor (ROR)-γt and are further classified as lymphoid tissue-induced (LTi) cells and natural cytotoxicity receptor (NCR)-negative cells or NCR-positive cells [10]. The diversity of ILC subgroups within groups is extensive, such as ILC1a-d, ILC2a-d, and ILC3a-e subgroups [11]. Regulatory ILCs (ILCregs), similar to regulatory T-cells (Tregs), were discovered in the intestine in, addition to these three conventional subgroups, secreting TGF-*β* and inhibiting ILC1s and ILC3s in an IL-10-dependent manner [10]. Circulating ILCs do not appear to fit into this scheme despite their phenotypic similarity to their tissue-resident counterparts; hence, they are not discussed further in this article [12].

Various organ-specific subpopulations of ILCs are found in multiple non-lymphoid tissues, including lung, skin, intestine, liver, and adipose tissue [13]. Intraepithelial ILC1s were enriched in the oral mucosa, and ILC2s were predominantly enriched in oral draining lymph nodes (dLNs). Compared to the draining mesenteric lymph nodes, ILC populations are more abundant in the lower GI tract’s mucosal surface [14]. The data reported here suggest that a similar phenomenon happens in the oral mucosa because the gingivae have an enrichment of ILCs and a more diverse ILC population than oral dLNs [15]. It is hypothesized that the increase in the variety and amount of ILCs in gingiva should react to ongoing microbial disturbance in the mouth to preserve oral homeostasis. In periodontal inflammatory tissues, the number and frequency of different ILC subpopulations increase, thereby promoting inflammation [15,16]. However, this phenomenon further validates the above speculation. The ratio of ILC2s and ILC3s in the gingiva of mice and humans differ significantly. Human gingiva only has a modest percentage of IL-17^+^ILC3, and no ILC2 population has been found [16]. The gingiva of mice has roughly similar amounts of ILC1 and ILC2 [15]. This variation may be due to environmental effects brought on by different microbes in different racial groups or between some non-pathogenic mice and humans.
ijms-24-06099-t001_Table 1Table 1Characteristics of innate lymphoid cells in mouse and human.ILC GroupSubsetsStimuliMediatorsPhenotypeTranscriptionFactorsReferencesMouseHumanGroup 1ILC1stumors, intracellularmicrobe(bacteria, virus, parasites)IFN-r, granzymes,perforinCD49a, NK1.1, NKp46, CD122, CD127, CD200R, TRAILNKp46, CD122, CD127, CD200R, TRAILT-betEOMES^hig^[17,18,19,20,21]NK cellsCD49b, NK1.1, NKp46, CD122, KLRG1NKp46, NKp30, NKp80, CD56, CD16, CD122, CD127, KLRG1T-betEOMES[20,22,23,24]Group 2ILC2sextracellular parasites and allergensIL-4, IL-5, IL-9, IL-13, AREGCD127, CD25, ST2, KLRG1NKp46, CD127GATA3RORα[25,26,27]Group 3NCR^+^ILC3sextracellular microbes (bacteria, fungi)IL-22, IL-17, GM-CSF, LymphotoxinNKp46, CD127NKp46, NKp44, CD127, CCR6, CD56RORγtAhR[9,28,29]NCR^−^ILC3sNKp46, NKp44, CD127, CD56, CCR6CD127, CCR6[30,31]Ltimesenchymal organizer cells(CXCL13, RANKL)RANK, Lymphotoxin,TNF, IL-17, IL-22CD127, CCR6CD127, CCR6, CD7[31,32]


## 3. The Development of ILCs

Mouse and human ILCs differ significantly in several key aspects, such as surface markers, phenotype, and several genes, including Id2, NFIL3, Zbtb16, GATA3, and Tox [10,15,33]. The developmental lineage of mouse ILCs is being explored using genetically modified animal models, but knowledge of human ILC development is very limited. The first reason for this is the lack of comparable genetic and tracing techniques in animal models of lymphoid development. The second is the variation in how progenitor cells are defined, lineage identification that is not reliant solely on cytokine production, and other indicators.

Similar to other lymphocytes, mouse ILCs originate from a common lymphoid progenitor (CLP), with ILCs predominantly present in the α4*β*7-positive portion of the CLP population (α4*β*7^+^ CLP). Downstream of the murine CLPs, two different progenitors have been discovered, each with limited ILC potential. These include common ILC precursors (CILPs) and early innate lymphoid progenitor cells (EILPs) [34,35,36]. It has also been suggested that CILP may serve as an intermediary step of maturation between CLP and EILCP given the significant decrease in CILPs, along with EILCPs in Tcf7-deficient mice [37]. EILPs have lower levels of Zbtb16, IL-7Rα, and Id2 than CILPs. Integrins α4*β*7 and CXCR6 are markers for CILPs. CILPs can differentiate into at least two different precursors: NK progenitors (NKPs), which can give rise to cNK cells; and common helper ILC progenitors (CHILPs), which can differentiate into all ILC subpopulations [38]. CHILPs express Id2 and CD127 and contain two groups: the group of cells that express PLZF (promyelocytic leukemia zinc finger), called ILC precursor cells (ILCps), and a population of RORγt+ PLZF− cells, which highly express TOX and not GATA3, called lymphoid tissue inducer precursors (LTips) [39]. LTips differentiate into LTi cells, and ILCps can further differentiate into ILC1s, ILC2s, and ILC3s, depending on the transcription factors T-bet, GATA-3, and RORγt [40]. Unfortunately, while it is understood that ILCregs do not originate from ILCps in the intestine tissue, the specific origin of ILCregs from CHILPs remains unknown [10] (Figure 1).

In human, early tonsillar progenitors (ETPs), which were once referred to as stage 1 and stage 2 NK cell developmental intermediates (NKDIs) and are currently detected in secondary lymphoid tissues (SLTs), are the earliest ILC progenitors that have been identified. Specific classifications of ETPs include Lin^−^CD34^+^CD10^+^CD117^−^ ETP1 and Lin-CD34^+^CD10^−^CD117^+^ ETP2 [33]. The expression of IL-1R1 is variable in ETP2s. IL-1R1+ ETP2s are ILC-restricted, characterized as ID2^+^RORγt^+^RAG1, while IL-1R1^−^ETP2s show some T-cell and DC potential [41]. IL-1R1^+^ ETP2s are the earliest committed human CILCP that has been found so far. Stage 3 NKID has restricted potential and ILC generation, which has the phenotypic characteristics of a Lin^−^CD34^−^CD7^+^CD127^+^CD117^+^CRTH2^−^ ILC progenitor (ILCp) [42]. The expression of NKp46, CD56, and killer cell lectin-like receptor subfamily G member 1 (KLRG1) can be used to distinguish between several ILCp populations with restricted differentiation potential. The ability of CD56-positive ILCps to differentiate into NK cells, ILC1s, and ILC3s is limited, whereas NKp46-positive ILCps mostly differentiate into ILC3s, and KLRG1-positive ILC precursors primarily develop into ILC2s [43,44] (Figure 2).

## 4. Plasticity of ILC Subsets 

Effective innate immune responses depend heavily on the heterogeneity and plasticity of ILCs. ILCs interact with epithelial cells, stromal cells, myeloid cells, and adaptive immune cells in the surrounding environment to modify their activities and phenotypes in response to environmental cues, known as plasticity. The phenotype of ILCs is relatively flexible because the local tissue microenvironment in inflamed tissue can differ from that of healthy tissue. ILC subsets are selectively activated and accumulate, and the equilibrium between cytokines and chemokines is frequently disturbed. Finally, a subset of ILCs with a new specialized function forms through a process termed as transdifferentiation. Transdifferentiation increases the immune system’s effectiveness because it avoids the need for progenitor cell differentiation.

In 2010, it was discovered, for the first time, that ILCs have plasticity comparable to T-cells in mice and humans [45,46,47]. Environmental cues can trigger ILCs to become increasingly plastic and interconvert their phenotypes, even if the phenotype of ILC subsets is well defined. ILCs precursors develop from CD34^+^ myeloid precursors to circulating naïve ILCs that exist in peripheral blood and undergo further maturation in tissue, known as tissue-resident ILCs. Before the terminal differentiation of ILC subsets, circulating naïve ILCs undergo a plasticity phase [48], and both extrinsic and intrinsic divers of ILCs plasticity have been documented [49].

In inflammation-related diseases, ILCs plasticity is a ubiquitous phenomenon involved in the regulation of immune response and inflammation. Based on the transcription factor RORγt, ILC1 differentiated to ILC3 in the presence of IL-2, IL-23, and IL-1*β*, and this process was accelerated in the presence of retinoic acid [50]. Furthermore, transdifferentiation is a reversible process. A middle ILC3-ILC1 cluster with a high directional preference for ILC1s in human tonsils and intestine was discovered by RNA velocity analysis [51]. Changes in the epigenetic landscape result in the transition of ILC2 into ILC1 in an IL-1*β*-receptor and IL-12-IL-12R pathway-dependent manner. Although their transdifferentiation in oral mucosal still needs to be explored, this occurrence is generally associated with Crohn’s disease and chronic obstructive pulmonary disease (COPD) [52,53,54]. Inflammatory ILC2 cells (iILC2 cells) can transform into ILC3-like cells and develop the capacity to generate IL-17, contributing to immunity to both fungi and helminths, during Th17 polarization circumstances in vitro or Candida albicans infection in vivo [55]. ILC2s exhibit phenotypic plasticity during the psoriasis inflammatory process. ScRNA-seq validated the dynamic changes in the status of the original or static ILC2, as well as the metastasis of the effector ILC2 [56]. In response to TNF-*α*, IL-1*β*, and IL-23, ILC2s transform into ILC3-like cells that produce IL-17. Meanwhile, IL-4 has the power to reverse this event [57]. This process has also been noted in nasal mucosal, and this transdifferentiation is abrogated by vitamin D3 and IL-4 [58]. However, further research is required to confirm this process in the oral mucosa. Notably, there is currently no published research on human ILC2-ILC3 plasticity. A subset of NCR^+^ ILC3s suppressed the conversion of RORγt into ILC1-like cells during oral microbial infection. ILC1-like cells are phenotypically and functionally similar to ILC1s, although they are not actual ILC1 subpopulations, and are hence referred to as “ex-RORγt ILCs”. ILC3s convert from RORγt^+^ ILC3 to ex-RORγt^+^ ILC3 in chronic inflammatory situations to adjust to environmental cytokines and preserve tissue homeostasis.

ILCs infiltrate the tumor microenvironment and respond rapidly to inhibit tumor growth. Transforming growth factor-*β* (TGF-*β*) and IFN-γ produced by ILC1s has an important role in the immune response to tumors. TGF-*β* signaling triggers the conversion of ILC3 to ILCreg by scRNA-seq to promote tumor growth [59]. TGF-*β* signaling also promotes the conversion of NK cells to intermediate ILC1s (intILC1s) to promote immune escape and tumorigenesis. NK cells transform into ILC1-like cells in TGF-*β*-rich tumor environment [60]. Oral squamous cell carcinoma (SqCC) is a major type of oral cancer. ILC1 to ILC3 conversion by SqCC tumor cells results in the production of IL-23, which encourages IL-17-mediated tumor cell proliferation [61]. 

The plasticity of ILCs is intimately correlated with the development of illness, and the development of disease can be somewhat managed by regulating cell plasticity. The immunological microenvironment is altered due to the illness process, although it is yet known whether the plasticity of ILCs is a cause or an effect of disease onset.

## 5. ILCs in Periodontal Homeostasis

The modulation of the activation state and quantity of local ILCs, which involves three main aspects, is what determines the quality and strength of the local immune response in the oral mucosa, as follows: (1) activation or inhibition of tissue-resident ILCs; (2) plasticity and differentiation of specific ILC subpopulations; and (3) tissue-specific migration and regional accumulation of peripheral ILCs [27]. It is significant to highlight that ILCs play a crucial function in the oral mucosa given that all three aspects of ILCs are pertinent to immune response. They are a potential therapeutic target for maintaining periodontal homeostasis because, while they play a protective role in periodontal tissues, they may also initiate and amplify locally detrimental immune projections (Figure 3).

Under the condition of infection or chronic inflammation, the number of ILCs on the local oral mucosal surface [62] and draining lymph nodes is significantly increased to effectively regulate the innate and adaptive immune response of oral microorganisms [63]. This population is partly supplemented by a small amount of circulating precursor cells and circulating mature ILCs [64], as well as the migration of specific subpopulations of ILCs from the mucosal surface in the presence of chemokines and homing receptors [14]. However, the mode and mechanism of ILC migration from the gingiva to the oral drainage lymph nodes remains to be investigated. Although the majority of oral draining lymph node and gingival populations are CD117^−^ NKp46^−^ ILC subpopulations, it has been shown that the gingival epithelium and oral draining lymph nodes produce INF-γ and IL-5 ILC subpopulations in close proportions [15]. This may be related to the potential plasticity and heterogeneity of ILCs.

### 5.1. ILC1

ILC1s are a heterogeneous mixed-cell population, a class of helper ILC1 (also known as tissue-resident ILC1), characterized as lin^−^CD127^+^CD117^−^CRTH2^−^NKp44^−^, with the ability to express T-bet and produce TNF-α and IFN-γ. It has been determined that human gingival tissue is dominated by the ILC1 subgroup [65], with the largest number of ILC1s in the periodontium and gingiva (approximately 50% of total ILC1s) [66]. In addition, a small subset of RANKL-expressing ILC1s was also found, which may play an immune regulatory role in gingivitis and periodontitis lesions [65]. 

Another classical member of this population is the NK cell, which expresses CD62L, CCR7, and S1PR, and highly expresses granzyme and perforin. NK cells have a distinct origin in comparison to ILC1. More in-depth studies utilizing polychromic receptor mice have revealed that ILC precursors, which were previously believed to be unable to develop into NK cells, exhibit significant NK cell precursor activity. As a result, the difference between ILC1s and NK cells is considerably less apparent because certain subsets have been described that combine ILC1 and NK cell characteristics. NK cells represent innate CD8^+^ CTL, which interacts directly with periodontopathogenic microorganisms [67], such as *Aggregatibacter actinomycetemcomitans*, *Porphyromonas gingivalis*, and *Fusobacterium nucleatum*. This interaction induces NK cells to form immune synapses, followed by polarization of their microtubule organizing centers (MTOCs) and secretory lysosomes towards the lysis synapses [68]. Prior to fusing with the plasma membrane, secretory lysosomes settle and increase IFN-c and TNF-α levels [69], ultimately leading to periodontal tissue destruction and alveolar bone resorption. In addition, NK cells have long been demonstrated to be crucial in the prevention of human and monkey immunodeficiency virus (HIV/SIV) by preventing viral multiplication and transmission in the oral cavity [70]. 

Helper ILC1 subpopulations and NK cells contribute to the IFN-γ-mediated proinflammatory release of IL-12, IL-15, and IL-18. NK cells share a large degree of similarity in their properties and functions with helper ILC1s, which also produce IFN-γ. IFN-γ is expressed at higher levels in periodontitis than in gingivitis, indicating that it may be implicated in the pathogenesis of oral mucosal inflammation through the aberrant secretion of IFN-γ, which induces chemokine production by epithelial cells. In the uninfected oral epithelium, ILC1s cross the basal epithelium to maintain barrier integrity and homeostasis. Numerous studies have shown that ILC1s can secrete IFN-γ to induce a local antiviral state during early phases of viral infection [69,71,72,73]. First, the source of IFN-γ is determined to clarify how helper ILC1s, rather than NK cells, play a role in restricting viral propagation using Rag2^−/−^ IL-2rg^−/−^ animals (lacking all ILC subgroups) or mice treated with NK1.1 (lack of NK cells and ILC1) [72]. Depletion of ILC1 during VACV infection and IFN-γ-mediated neutralization of viral load in uninfected tissues provide additional evidence that ILC1-mediated antiviral effects depend on steady state or early secreted IFN-γ during VACV infection. IFN-γ-regulating genes (IRGs), including IRF7, were significantly upregulated in the oral mucosa following viral infection [69,72,73]. Prior to viral infection and near to the ILC1 area, IRF7 was expressed concentrically in the uninfected mucosa [71]. It has certainly been demonstrated that ILC1s secrete INF-γ to create a local antiviral state in the oral mucosa before viral infection in order to stop viral reproduction and transmission after infection has occurred. In addition, IFN-γ is associated with bone through a complex and context-dependent mechanism and is thought to regulate osteoclast differentiation and bone resorption in periodontitis.

### 5.2. ILC2

The ILC2 subpopulation highly expresses transcription factor GATA3 and is characterized as lin^−^KLRG1^+^IL-7R^+^CD117^−^IL33R^+^ IL1R2^+^. Two functionally and phenotypically unique subpopulations of ILC2 in mice have been identified. Inflammatory ILC2s (iILC2s), which are exclusively produced by IL-25 activation in an inflammatory state, and natural ILC2s (nILC2s) are defined as a homeostatic and IL-33-responsive cell [55]. There is still no published evidence that ILC2 subpopulations exist in humans, and single-cell RNA sequencing was unable to identify transcriptionally different subpopulations of human tonsillar ILC2 [74]. 

ILC2s are distinguished by the expression of transcription factor Bcl11b, which regulates mature ILC2s’ identity and function [75], the IL-33 receptor, prostaglandin D2 receptor 2 (CRTH2), and by varied levels of c-Kit, all of which are important for ILC2s’ location and function. ILC2s can also be identified by the expression of the killer cell lectin-like receptor subfamily G member 1 (KLRG1), a co-inhibitory receptor that has previously been discovered on T and NK cells [76]. KLRG1 is increased in response to IL-25 during infection and binds to members of the cadherin family [55].

ILC2s participate in several functions, such as lipid metabolism, parasite defense, and accumulation during type 2 inflammation. A previous study showed that ILC2s had a significant effect on periodontitis. In a ligation-induced periodontitis mouse model, AMPK knockdown promoted the release of IL-33 from ILC2s, a novel compensatory mechanism for suppressing inflammation [62]. ILC2s have been attributed to the regulation of inflammatory and immune responses by IL-33 in asthma, allergic dermatitis, and allergic lung inflammation [77,78]. The synthesis of ILC2s, which secrete substantial levels of IL-5 and IL-13, can be activated and encouraged by IL-33 derived from epithelial cells. IL-33 promotes STAT3 translocation into mitochondria via MAPK-induced STAT3 phosphorylation at the S727 site. This, in turn, controls electron transport chain activity and cellular respiration and lessens ILC2-mediated inflammation in acute asthmatic lungs [79]. In contrast, IL-33 promotes ILC2 growth and activation, and activated ILC2s recruit adaptive immune cells through high expression of CCL5 to exert anti-tumor immune effects [80]. 

### 5.3. ILC3

ILC3s are the most diverse mimics of RORγt-dependent Th17 cells available, including LTi cells. They are characterized as lin^−^CD127^+^CD117^+^ and express transcription factors RORγt and AHR [9]. NKp46, NKG20, and NK1.1 are three RORγt NCRs found in the ILC3 population. ILC3s are finely separated into NCR^+^ ILC3 (4%) and NCR^−^ ILC3 (18%) in periodontal inflammatory tissues [65]. While NCR^−^ ILC3s primarily produce IL-17A, NCR^+^ ILC3s can produce IL-17A, IL-22, and IFN-γ [81]. NCR^+^ ILC3s are the exclusive source of IL-22 in adult tonsils. ILC3s are the primary source of IL-17A in periodontal inflammation and fungal infection, despite the fact that Th17 cells are the primary source of IL-17A in other tissues [17]. In addition to these two subpopulations, two other subpopulations of ILC3s have been identified in human tonsils: HLA-DR^+^ILC3, which account for approximately 10% of all ILC3s in tonsils; and CD62L+ILC3, which express CD45RA but do not secrete IL-22 and IL-17, and are therefore presumed to be naïve ILC3s [74,82]. ILC3s are under-represented in the pool of circulating lymphocytes, as opposed to tissues. In contrast to mature ILC3s seen in secondary lymphoid organs, the majority of circulating ILC3s exhibit low amounts of RORγt. Circulating ILC3s are multipotent ILC precursors (ILCPs), which nonetheless have the capacity to give birth to functionally mature helper ILC subsets and NK cells.

Lymphoid tissue inducer (LTi) cells are regarded as belonging to a distinct ILC lineage. When the ILC3 group was first described in 1997 [83], LTi cells were its first members. In the developing fetus, LTi cells contribute to the organogenesis of secondary lymphoid tissues. In fact, LTi cells produce IL-22, IL-17A, and IL-17F as a result of RORγt expression and are involved in the creation of secondary lymphoid organs during embryogenesis [84] and maturity, as well as in their restoration after infection [85]. LTi cells in the mouse develop in a manner distinct from helper ILC3 cells. However, it has been challenging to identify a specific LTi population in humans. In humans, NRP1+ ILC3 generates substantial amounts of IL-22 and IL-17A [86]. These cells could therefore be a representation of LTi cells in humans. It is yet uncertain, though, whether these cells develop differently from helper ILC3s than they do in mice.

The dual role of ILC3s depends on cytokine levels. Intestinal epithelial cells are induced by IL-22 to produce antimicrobial peptides quickly, such as α- and *β*- defensins, and IL-22 also enhances epithelial cell renewal to maintain epithelial barrier integrity. IL-17A and IFN-γ trigger the pathogenesis of periodontitis by inducing TNF-α and RANKL production to induce periodontal tissue destruction [17]. However, ILC3s reduce the production of IFN-γ and IL-17 by decreasing CCR6 expression in response to IL-23 stimulation while increasing IL-22 production [87], thereby suppressing inflammatory response. In addition, ILC3s may limit the multiplication of periodontal microorganisms and their toxic products to avoid entering the systemic immune system and causing systemic diseases, such as Alzheimer’s disease. It is possible that ILCs encourage endostasis by promoting the development of isolated lymphoid follicles and epithelial cell foci with the aid of dendritic cells and Tregs, or by triggering the production of GM-CSF through the flora.

## 6. Conclusions

The developmental lineage and characterization of ILCs has provided new insights into immune response in periodontal tissues. This new group of innate immune cells in periodontal tissues contributes to resistance to pathogen invasion and viral replication, promotes local oral tissue repair and inflammation, regulates periodontal tissue homeostasis, and maintains the integrity of the oral mucosal barrier. However, further investigation of the molecular mechanisms that make these processes possible is required.

ILCs can adjust to shifting periodontal tissue circumstances owing to their plasticity, which may be crucial for fine-tuning responses to various pathologic stimuli. Given that the majority of canonical ILC subsets are tissue resident under homeostatic conditions, their plasticity enables an immediate response to alterations in the microenvironment brought on by pathogens without the requirement for de novo recruitment of ILC subsets. Restoration of the original ILC subset composition may occur along with the resolution of inflammation. Future research should pay close attention to the unraveling of metabolic programs and signaling cascades that control ILC differentiation and plasticity. It may be possible to better understand the role that ILCs play in maintaining periodontal homeostasis by pursuing further research on ILC heterogeneity and plasticity, which may also lead to the discovery of new biomarkers and therapeutic targets for cutting-edge, individualized treatments.

## Figures and Tables

**Figure 1 ijms-24-06099-f001:**
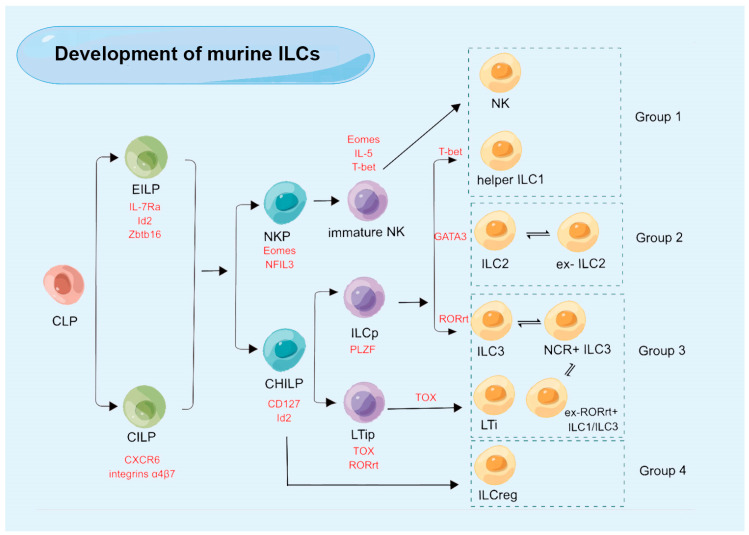
The development of murine innate lymphoid cells (ILCs). CLP: common lymphoid progenitor; EILP: early innate lymphoid progenitor; CILP: common ILC precursor; CHILP: common helper ILC progenitor; NKP: natural killer progenitor; ILCp: ILC precursor; LTip: lymphoid tissue inducer precursor; LT: lymphoid tissue inducer; ILCreg: regulatory ILC.

**Figure 2 ijms-24-06099-f002:**
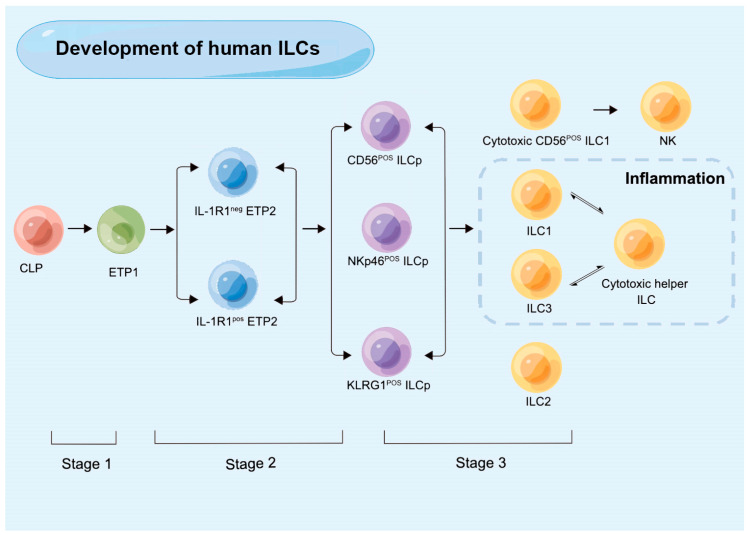
The development of human innate lymphoid cells (ILCs). CLP: common lymphoid progenitor; ETP: early tonsillar progenitors; ILCp: ILC precursor; KLRG: killer cell lectin-like receptor subfamily G member 1.

**Figure 3 ijms-24-06099-f003:**
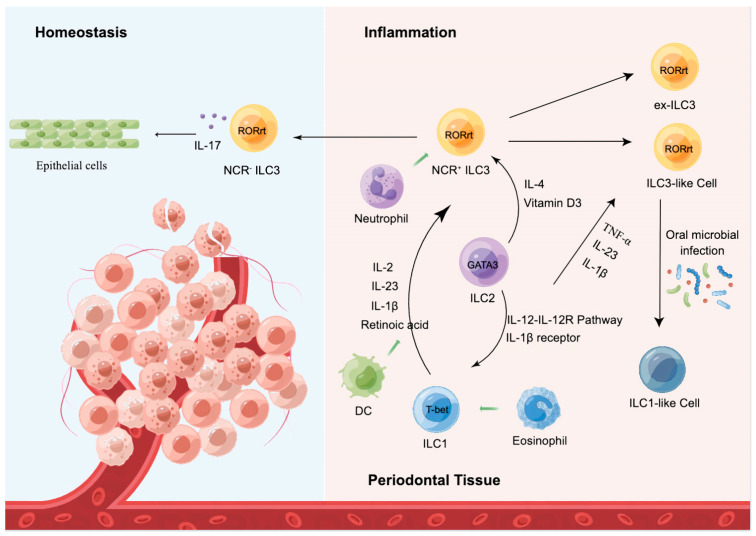
ILC subsets maintain periodontal homeostasis. At steady state, ILC subsets reside within the basal layers of the oral mucosal epithelium, contacting the basement membrane and slowly moving across the mucosal epithelium via contacting and squeezing between mucosal epithelial cells. At inflamed state, they generate cytokines and interact with other immune cells, such as neutrophils and eosinophils, to control ILCs’ plasticity during homeostasis via a complex network of transcription factors, resulting in the establishment of an antiviral/inflamed state of the mucosal epithelium.

## Data Availability

Not applicable.

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
