# Peer review of "Function of Innate Lymphoid Cells in Periodontal Tissue Homeostasis: A Narrative Review"

_ijms, 2023, doi:10.3390/ijms24076099_

Round 1

Reviewer 1 Report

This study is about the function of innate lymphoid cells on homeostasis in periodontal tissue. The topic is clinically important and interesting for the readers of the IJMS. After some minor revisions, I will strongly recommend publishing this manuscript for IJMS. My comments are as below.

#1. Abstract

The current version of the abstract looks like just referring general information and the purpose of this study. The abstract should include not only the introduction and purpose of this study but also the main contents and conclusion.

#2. (P1L41) "ILCs divided into five distinct ILC subsets, namely helper ILCs (i.e., ILC1s, ILC2s, ILC3s), natural killer (NK) cells, and lymphoid tissue inducer (LTi) cells"

 "divided" should be replaced with "are divided".

#3. (P2L59) There should be some description(s) about the purpose of this review at the end of the last paragraph in Introduction section.

#4. (P2L73) "futher" should be "further".

#5. (P2) Table 1

Authors should clearly refer to Table 1 in the text. Additionally, in the row "ILC Group", I recommend removing "ILCs" from each group, and describing just like "Group 1", "Group 2", and "Group 3".

#6 (P3L81) "dLNs. [8]." should be "dLNs [8]."

#7. (P3L120) ". (Figure 1)" should be " (Figure 1)." (the position of “.” Is incorrect).

#8. (P7L262) "mucosa[49]" needs a space between "mucosa" and "[49]".

#9. (P7L277) "[53] ," the space between "[53]" and "," is not needed.

Reviewer 2 Report

Zhiyu Ma et al presents an interesting review article surrounding the mechanisms of ILCs in periodontal tissues and the host factors to provide new strategies for the treatment of periodontitis.

This manuscript is of general interest to the readership of IJMS Journal

Minor consideration: 

1. Inclusion of a schematic diagram of multifaceted roles of ILCs subsets in periodontal tissues would help highlight the central message of the paper.

2. The roles of ILCs subsets in periodontal tissues should be discussed and elaborated more. I feel that these roles are modestly discussed.

Reviewer 3 Report

In title there should be a mention that this is a narrative review 

The abstract should be more elaborate.

There Introduction is very concise and has only 1 reference. The authors should address this lacunae in literature with references.

Please consider writing as conclusion and not remarks. 

Also having a table of the studies included in each sections will be easier to the readers and to reproduce or update it. 

Round 2

Reviewer 3 Report

Congratulations to the authors for addressing all my concerns satisfactorily 

I see this paper flying high soon